# Web Geographic Information System: A Support Tool for the Study, Evaluation, and Monitoring of Foci of Malaria Transmission in Mexico

**DOI:** 10.3390/ijerph20043282

**Published:** 2023-02-13

**Authors:** René Santos-Luna, Susana Román-Pérez, Gerardo Reyes-Cabrera, María del Rosario Sánchez-Arcos, Fabián Correa-Morales, Marco Antonio Pérez-Solano

**Affiliations:** 1Subdirectorate of Medical Geography and Geomatics, National Institute of Public Health, Cuernavaca 62550, Mexico; 2Subdirectorate of Vectors, National Center for Preventive Programs and Disease Control, Mexico City 06100, Mexico

**Keywords:** GIS-based web information system, malaria elimination, entomological surveillance, vector control, dashboard

## Abstract

Malaria is currently an endemic disease in Mexico. The country joined the WHO’s E-25 initiative for the elimination of Plasmodium vivax to achieve elimination and certification within the established period. Having a Web-based information system was, therefore, deemed necessary to assist in the detection, investigation, and elimination of transmission in the foci, as well as for the timely treatment of malaria-positive cases. The “Information System for the Elimination of Malaria in Mexico” was designed, developed, and implemented with a geographic vision, which includes a Web tool to georeference homes and aquatic systems, a dashboard and an indicator evaluation card for monitoring activities, notification of probable cases, and vector control among other indicators. The implementation of the system was gradual in the seven states that are currently in the malaria elimination phase; subsequently, the system was implemented in non-transmission states. In 2020, the system implementation stage began; first, the basic data of more than 96,000 homes throughout the country were georeferenced, and then the primary data capture tools of 17 formats, 32 reports, and 2 geographic viewers were enabled for information queries. A total of 56 active foci have been identified in 406 localities as well as 71 residual foci in 320 localities. Recently, the Foci Manager was developed, which is a specific tool for the study, evaluation, and monitoring of active foci through a GIS, a dashboard, and a systematized evaluation certificate. Georeferencing tools decreased the cost of spatial data collection.

## 1. Introduction

Malaria is a disease caused by parasites of the genus *Plasmodium vivax* and *falciparum*. It is transmitted by female mosquitoes of the genus *Anopheles* and the species *pseudopunctipennis*, *albimanus*, *vestitipennis*, *darlingi*, and *punctimacula*. In 2021, it was estimated that there were more than 247 million cases of malaria in the world, with Africa accounting for 95% of all cases. In the region of the Americas, the WHO estimated that there were more than 597,000 cases [1]; in addition to presenting a lower number of cases, there was also a decrease in the geographical coverage of regions with local transmission. In Mexico in 2020, 339 confirmed cases were recorded, according to the epidemiological bulletin of the National Epidemiological Surveillance System (SINAVE) [2]. In recent decades, several strategies have been implemented in Mexico, and each one has had a significant impact on reducing malaria cases. Currently, transmission continues in seven of Mexico’s 32 states (Figure 1).

Mexico joined the E-25 Malaria Elimination Initiative, which establishes a commitment to the Pan American Health Organization (PAHO) to achieve malaria-free status by 2025. The Federal Technical Group for the Elimination of Malaria, which is responsible for guiding the strategy and achieving the goal, is composed of experts from four institutions: the General Directorate of Epidemiology (DGE), the Institute of Epidemiological Diagnosis and Reference (InDRE), the General Directorate of Health Promotion (DGPS), and the National Center for Preventive Programs and Disease Control (CENAPRECE). The InDRE is the institution that coordinates the Microscopy Network; this network receives and analyzes the samples from active, passive, or reactive case surveillance. Positive results are notified to the DGE and to the CENAPRECE malaria program, which is in charge of providing treatment to the patient. The DGE is the governing body of the National Epidemiological Surveillance System (SINAVE). The SINAVE is the official source of information on confirmed cases of malaria. CENAPRECE is the institution that coordinates the activities of the National Vector Control Program, among which is the active, passive, and reactive search for probable cases of malaria. In the transmission areas, its brigades routinely tour the locations of active foci, and it participates in activities to treat confirmed cases. The National Vector Control Program is also responsible for performing entomological surveillance and carrying out vector control actions in Mexico.

The Global Malaria Programme (GMP), through its “*Malaria surveillance, monitoring & evaluation: A reference manual*” publication, defines “that surveillance consists of the continuous and systematic collection, analysis and interpretation of disease data, and the use of this information in the planning, implementation and evaluation of the exercise of public health” [3]. This highlights the importance of information systems as a fundamental source for the collection, processing, and interpretation of data.

The Global Technical Strategy for malaria 2016–2030 suggests transforming surveillance into a core intervention, where the objectives are to detect, research, and eliminate the sources of continuous transmission [3]; the strategy also emphasizes that information systems are essential tools for surveillance.

China’s malaria control program designed and implemented a communication and information system for the surveillance and monitoring of response activities for malaria elimination in 2012 [4]; four years later, China recorded its last malaria case. In 2021, China was certified malaria-free by the WHO. This successful case encourages each country to develop and adapt a tool to their needs and conditions. Another successful example is from Thailand in 2009, where the Center of Excellence in Biomedical and Public Health Informatics (BIOPHICS) and the University of Mahidollogró developed an Electronic Malaria Information System (eMIS). This system was evaluated nine years after its implementation in order to improve the quality of the malaria surveillance system [5]. The WHO recognizes the importance of having a computer platform for real-time reporting [6]. This serves as an example that after a time of implementation of a system, it is important to carry out evaluations that denote areas of opportunity to improve the information systems.

In 2013, in Mexico, the national malaria program had a Web-based information system that was developed and operated by a software development company. This system was characterized as being an expensive system, with data capture functions in general formats without official identification codes. This fact made it impossible to relate to official databases such as the demographic or geographic databases of the National Institute of Statistics and Geography (INEGI) or those of SINAVE of the General Directorate of Health Information (DGIS). These data sources are of paramount importance for the malaria elimination strategy.

In 2019, the Subdirectorate of Medical Geography and Geomatics (SGMG) of the National Institute of Public Health (INSP), in coordination with the National Center for Preventive Programs and Disease Control, proposed a new version of the information subsystem to strengthen the activities of the national vector control program. CENAPRECE and INSP have relevant experience from implementing the Web Geographic Information System for Entomological Surveillance and Dengue Control [7]. Subsequently, they jointly developed specific information subsystems to manage the inputs (insecticides, vehicles, and application equipment), monitor species of arthropods of medical importance, and register operating personnel. This set of information subsystems makes up the Comprehensive Vector Monitoring System (SIMV).

## 2. Methods

The extreme programming (XP) methodology was used to develop the “Geographic Information System, as Tool in the Elimination of Malaria” (GISTEM), which consisted of six phases (Figure 2). The main actors of the malaria program at the three levels of users (federal, state, and jurisdictional) were interviewed about the process of data flows and ways of using the information. A jurisdiction with “active foci” was also visited to observe the logistics of the fieldwork and to learn about the computer skills of the personnel and the installed infrastructure capacity. Seventeen primary data record formats were identified, and an item was added to the forms to incorporate the official encoding of the INEGI geographic code, the Unique Codes of Health Establishments (CLUES), and the Federal Taxpayers Registry Code (RFC) as personal identification. It was identified that brigade personnel use field sketches to locate homes, notification posts, and confirmed cases, marking vector control interventions both in homes and in aquatic systems. As a result of the process of interviews, field visits, and formats analysis, a software requirements specification document (vision document) was prepared. This document and the “*Malaria Surveillance, Monitoring & Evaluation: A Reference Manual”* were the basic tools for the conceptual design of the information system [3].

### 2.1. Database

The database is relational and allows storing, organizing, and querying spatial and tabular data according to the event (case, search, identification, diagnosis, notification, treatment, entomological study, or control action), space (focus, locality, dwelling, health center, or notification post), date (day, week, quarter, or year) and person (notifier, patient, or brigade member).

### 2.2. Conceptual Design and Development

A modular information system was designed that has data recording, georeferencing tools for dwellings, and tools for the consultation of tabular data through reports, dashboards, and geographical displays for the consultation of spatial data (Figure 3). The system modules were developed with Visual Studio.NET 2017, Microsoft SQL Server 2019 was used as the database administrator, and ArcGIS Server 10.6 architecture was used to publish the Web map services [8]. The geographical viewers and georeferencers were programmed using Visual Studio.NET 2017 with ArcGIS API for JavaScript 4.24.

### 2.3. Testing

To verify that the software complied with the expected functions, we carried out functional tests based on methods recommended by the International Committee for Software Testing Certifications (ISQTB), which were documented according to the IEEE 829 standard [9].

### 2.4. Implementation

In July 2020, the implementation phase of the system began in seven states; in September 2020, the implementation of the GISTEM began in the remaining 25 states, where only entomological surveillance activities were carried out. Implementation took place in three stages. The first stage consisted of training users to manage the “*Validation of locations”* geographical tool, which serves to validate the name and official geographical identification code and the spatial location of each location that belongs to a focus. In the second stage, users were trained in the management of the geographical tool “*Georeferencing of Houses* and *Aquatic Systems*” to locate and assign the identification of each house (ID-CNEP) on a Web map (Figure 4). The aquatic systems were subsequently georeferenced; with this tool, it was possible to transfer the information from the field sketches to a database. The third stage consisted of training the technical staff to record the data of the components of “Detection, Diagnosis and Treatment”, “Entomological Surveillance”, and “Control Actions” systems.

## 3. Results

The information system for the elimination of malaria can be found on the Comprehensive vector monitoring system portal http://geosis.mx/aplicaciones/sismv/ (accessed on 1 February 2023).

### 3.1. Routine Data Entry

In the detection, diagnosis, and treatment component, there is information on the activities that the brigade carries out in relation to the detection and active or reactive notification of cases, training for the reporting population, and detailed registration of the treatment administered to positive cases of malaria. This component has three data record formats: (a) service activities log (B1); (b) notification of probable cases (N1); (c) treatment record log (T1). The entomological surveillance component contains six data-recording forms for these activities: (a) recording of adult *Anopheles* with human bait in housing; (b) recording of adult *Anopheles* in a natural shelter; (c) recording of adult *Anopheles* in sprayed and unsprayed housing; (d) characterization of aquatic systems; (e) study of larvae and pupae; (f) biological testing. The third component was called “Control Actions”; in this component, all the chemical and environmental control activities that are carried out in the house, aquatic systems, and natural shelters are recorded. This component includes the following formats: (a) residual intra-domiciliary spraying; (b) flag delivery registration; (c) evaluation of the use of pavilions; (d) environmental nebulization activity; (e) elimination and modification activities of hatcheries and habitats of *Anophelinos* (EMHCA), with community participation; (f) application of larvicides; (g) control of adult *Anopheles* in natural shelters.

### 3.2. Data Output

The Malaria Elimination Information System generates 32 different reports: 5 for the detection, diagnosis and treatment component, 11 for the entomological surveillance component, and 16 for the control actions component. Users can generate reports with grouped (summation) or individual (nominal) data, with geographic coverage from their level of responsibility to the locality level. Reports are generated by date, epidemiological week, month, quarter, or year (Figure 5).

### 3.3. Geographic Viewer

The GISTEM spatial data output is accomplished through a geographic Web information system. The geographic viewer (GV) contains a query panel, so users can structure queries to see the characteristics of a focus for a period of time (quarter) and the location and environment of populations, roads, and orography, among other physical features. This is possible thanks to a mosaic of Web satellite images developed by the Environmental Systems Research Institute (ESRI). Users can also see the homes in each location and the aquatic systems of the environment duly identified and characterized. The GV shows the information that was originally in the field sketches; therefore, it is very useful for planning tasks. The consultation panel allows users to view homes with Rapid Diagnostic Tests (RDT), probable cases, treatments, and control actions. It also shows the entomological information and control actions for aquatic systems per quarter; additionally, the GV has a direct query function on the map elements; that is, with one click, a pop-up window with individual data is displayed. For example, if the query is about actions, the pop-up window contains, in addition to the geographical location and dates of the intervention, the type of insecticide used, characteristics of the application equipment, fuel consumed, number of rooms, and annexed rooms. If the aquatic system has not been worked, the reason why it was not worked is shown. Figure 6 shows a view of Güillachapa town that belongs to the “Cajón de Cancio” focus. This town is located on the edge of the Choix River; it has 51 homes and 4 aquatic systems. This figure shows the location of the dwellings: the red diamonds indicate probable cases, the dwellings where control actions were carried out through “intra-domiciliary spraying” are marked with green diamonds, and the dwellings without control actions are marked with yellow diamonds.

For aquatic systems, the dimension, use of water (irrigation, fishing, trough, etc.), characteristics of vegetation, and control interventions carried out in the current quarter are contained in the pop-up window. The GV is also useful for decision-making tasks.

### 3.4. Foci Manager

The GISTEM has an additional module called focus manager (GF); this module includes three basic tools. The first tool is used to define quarterly goals for the localities that make up a focus, which serves as a reference to evaluate compliance with the actions within the quarter. The second tool is the evaluation sheet called “Compliance evaluation indicators in the malaria elimination process in active foci”, which includes the activities of “Proactive case detection and notification promotion”, “Vector control”, “Entomological surveillance”, and “Treatment of cases and reactive response”. The evaluation card contains 57 indicators where global and individual activities are evaluated when it comes to first or second home visits, interventions, and post-interventions. Assessments can be generated quarterly; in addition, the national coordinator can make comments, indications, or recommendations after reviewing the evaluation cards. State and jurisdictional managers can download the evaluation sheet after it has been generated and reviewed by the national coordinator. The last tool of the GF is the board of indicators for the evaluation of the progress in the fulfillment of the activities during the current quarter. The dashboard shows the cumulative compliance of the activities during the current quarter, contains the 57 indicators of the evaluation card and three indicators of the resources available in the health jurisdiction: (1) quantity and types of insecticides in the warehouse; (2) vehicles and application equipment; and (3) personnel assigned to the malaria program. The dashboard graphically displays the overall progress percentages and the progress percentages for each indicator, updating as activities and cases are captured (Figure 7).

### 3.5. Data Recorded

Two years after the implementation of the information system, the program has 191 active users nationwide that have georeferenced more than 96,000 homes located in 1292 localities; in these homes, 251,110 probable cases (N1) have been presented, of which 8.19% were detected with RDT, with 289 tests being positive, which represents 1.40%.

In the last year, 11,225 RDT (Bioline MALARIA Ag Pf/Pf/Pv) were used for the detection of *P. falciparum* and *P. vivax*; Chiapas, Sinaloa, and Campeche are the states that have used the most RDT: 6533, 3917 and 709, respectively. However, Campeche is the state with the most positive RDT, with 12 tests. In these states, complete and timely treatment was provided, that is, within 72 h after the onset of symptoms for 84.21% of the patients who were detected.

Overall, 63.49% of the probable cases were detected by active search, 34.11% were detected by passive search, and 1.02% were detected by reactive search. In addition, 56 active foci with 406 locations and 71 residual foci with 320 locations were monitored during 2022.

The GV shows that, in the state of Sinaloa, the active transmission zone is currently to the north of sanitary jurisdiction number 1, which is located in the northwest of the state, bordering the states of Chihuahua and Sonora. In this area, 10 foci are being monitored: 2 active and 8 residual; in these foci, there are 60 localities, 7109 homes, and 16,878 inhabitants (Figure 8).

## 4. Discussion

In accordance with the provisions of the Global Technical Strategy against Malaria 2016–2030, it is necessary to invest in ordinary information systems, given the importance at any stage in the surveillance and supervision of the activities of antimalarial programs [10]. This highlights the importance and usefulness of information systems in providing timely information for decision-making.

Information and communication technologies today are applied in various areas, and public health is no exception. In this regard, it has been seen that in terms of public institutions, the development of systems requires very specific requirements, which are the joint responsibility of the key actors; that is, it is a shared responsibility between the person who designs and develops the system, and the users who use it [11]. In addition to this, the correct implementation of information and communication technologies in health systems requires taking care of some factors [12,13,14]. The human factor is one of them; in this sense, two essential elements can be highlighted in this project. The first element was to involve the key users, the national coordinators who contributed ideas to define the name of the system, and those who, at the state and jurisdictional level, proposed data output formats while other users participated in the pilot test. The participation of the users in the development of the system generated a sense of belonging. The second element was to create an identification logo for the system; the symbol represents the silhouette of a brigade member combined with the abstract shape of the *Plasmodium* parasite, acknowledging the control that operating personnel exerts over the parasite during this elimination phase. These two elements were fundamental in the implementation stage. Other important factors are technological and organizational leadership; currently, all jurisdictional headquarters have the technological communication infrastructure necessary for the proper functioning of the system, and in relation to the leadership and coordination of the national vector control program, they were clearly adequate during the system implementation phase.

The GISTEM is very useful in areas of malaria transmission in Mexico, where it is necessary to follow up the notification of cases with the investigation, classification, treatment, and control actions. However, it is also very useful in areas without transmission where surveillance is focused on identifying potentially receptive areas. This speaks of the adaptability that information systems must have. The GISTEM also stands out for being a simple and user-friendly system. It should be noted that the response from the users of the system regarding its use and acceptance was favorable and immediate. This occurs because of the leadership of the national management that has promoted and demonstrated the usefulness and simplicity of the system.

Building the information system from a geocoded database was essential to lay the foundation for the underlying georeferencing tools. Georeferencing web tools designed and developed for this project made it possible to transfer detailed data from hand-drawn maps to a central database. The “Domiciliary georeferencing” and “Georeferencing of aquatic systems” tools were designed to be operated from the jurisdictional office. The tools are friendly and allow the digitization of aquatic systems in rural locations. Generally, address geocoding services are used in housing georeferencing processes; however, to georeference homes in rural locations without GPS cartography, mobile devices must be used [15,16]. This processing option involves the purchase of GPS devices and the transfer of personnel to the field to obtain the coordinates of each home. In addition, an additional process is required for data integration and validation [17]. Instead, by developing the web georeferencing tools in this project, the processing time and cost of spatial data production were reduced. The use of official geocoding catalogs allowed us to relate our own databases with the spatial databases available from other official data-generating institutions to calculate the control panel indicators.

Since the implementation of the system, it has been possible to identify the homes where the cases are found, allowing the monitoring of patient treatment, as well as focusing entomological surveillance and vector control actions more effectively. This is reaffirmed in La Gomera in the Department of Escuintla, a municipality with a high incidence of cases in Honduras, where through microstratification and microplanning, it was possible to organize human resources more effectively [18]. To achieve this, it is of the utmost importance to be certain of the financial, human, and technological resources with which they are available, and all this is in the GISTEM.

As malaria transmission decreases, it becomes concentrated in specific areas, and the frequency of reporting increases; however, the use of information also increases due to the need for monitoring and evaluation of program activities [3]. A focus manager is a tool that calculates compliance assessment indicators of the operational activities in the active focus. The decision maker in the health jurisdiction can plan their control activities based on the information from the GF in order to determine specific courses of action. Dashboards always allow the most important information for decision-making to be observed quickly and easily. This is the case of the NIMR-MDB in India, which allows the visualization of malaria epidemiological data [19]. Through this dashboard, researchers and policymakers can perform epidemiological data analysis for the development of malaria control strategies in India.

The central process of the malaria elimination strategy lies in the opportunity to start the patient’s treatment 72 h after the date of onset of symptoms. The period for the detection and diagnosis processes can be very short: the sample must be transferred to the laboratory, then the result is awaited, and the patient is provided with treatment if it is positive. We know that malaria persists in rural areas of Mexico, which are generally dispersed and not very accessible, so the orographic conditions make transfers take a long time; in addition, figures from the 2020 Population and Housing Census revealed that 46.60% of rural localities in Mexico (with less than 2.500 inhabitants) do not have cellular signal coverage. This situation makes it impossible to capture a probable case in real time; for this reason, the data entry corresponding to the probable case notification is made until the laboratory result is obtained. In Mexico, this limits the ability to use mobile applications such as “The Malaria System MicroApp”, which is an automated diagnostic system to identify, through images, the species of *Plasmodium falciparum* in the development stage. The application works on mobile devices, and this makes it possible to overcome the accessibility barrier and minimize the time to start treatment [20]. There are other similar applications for the detection and classification of malaria parasite species [21,22,23]; however, given the conditions in Mexico, this option is not currently viable.

## 5. Scope

An important aspect to study in order to incorporate it into the GISTEM would be the possibility of using mobile applications for the detection of *Plasmodium* species; this would help a lot in dispersed and inaccessible rural areas. Finally, it would be important to evaluate the effectiveness of GISTEM in terms of user perception and to evaluate the results of the system to determine the impact and real contribution to the National Vector Control Program. This is in order to recommend the strategy implemented in Mexico to other countries.

## 6. Conclusions

The design, development, and implementation of GISTEM in Mexico is an experience that is desired to be shared, given the many complexities and vicissitudes encountered and overcome along the way. Taking shared responsibility between those who design and develop an information system and its users is key to understanding the needs and opportunities that can be offered to users. In addition, this makes the process of implementing the system easier and faster since it already has user acceptance at all hierarchical levels. Georeferenced malaria information can support transmission risk stratification to monitor patient treatment, as well as focus entomological surveillance and vector control activities. This experience will help generate evidence indicators for systematized surveillance and monitor the malaria certification process at both the regional and national levels.

## Figures and Tables

**Figure 1 ijerph-20-03282-f001:**
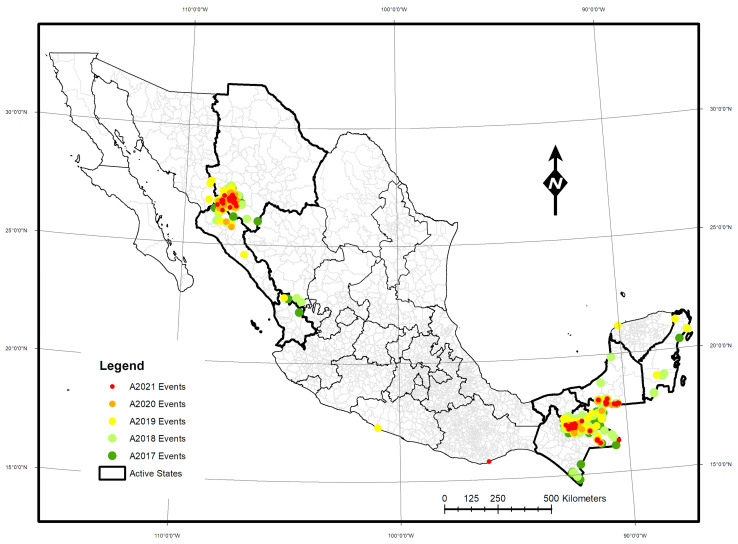
Confirmed cases of malaria from 2017 to 2021.

**Figure 2 ijerph-20-03282-f002:**
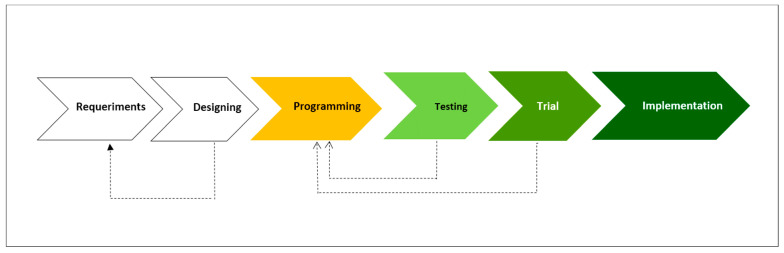
Phases of development, testing and implementation of the information system for the elimination of malaria.

**Figure 3 ijerph-20-03282-f003:**
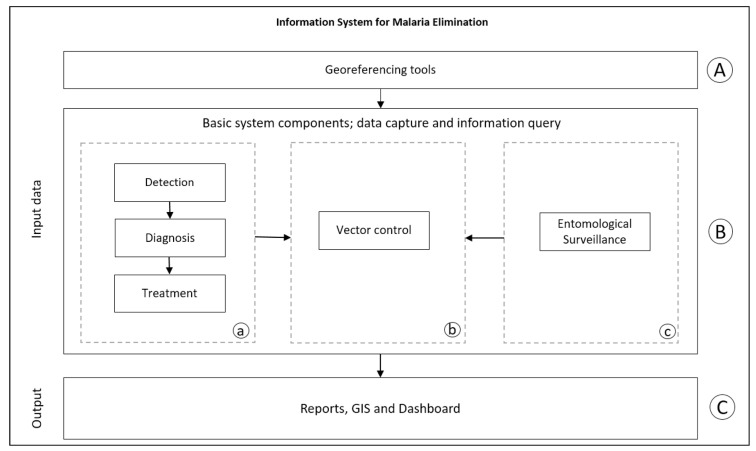
GISTEM modules and data flow. (A–C) Are the modules of system; (a–c) Are the componets of system..

**Figure 4 ijerph-20-03282-f004:**
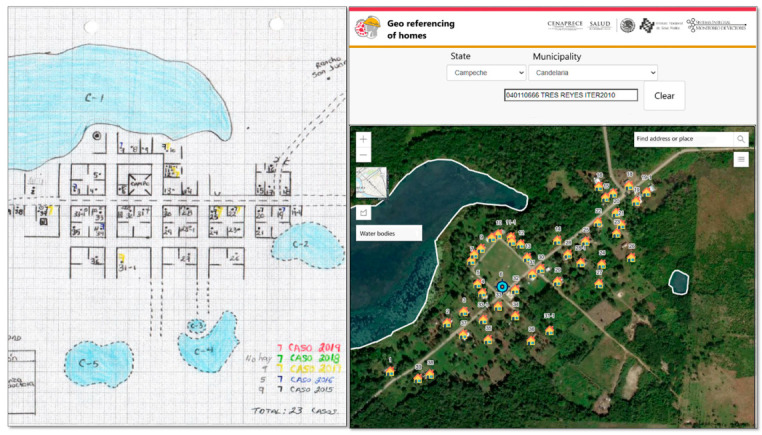
Georeferencing tool for dwellings and aquatic systems. In the houses the identification number is shown and in the white polygons the aquatic systems are shown. These elements can be related to epidemiological, entomological and vector control indicators.

**Figure 5 ijerph-20-03282-f005:**
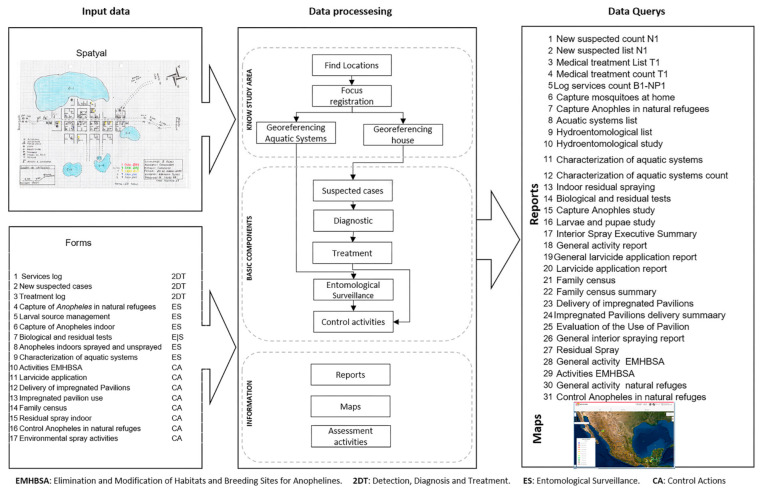
Data input, process, and output scheme. The input data contains 17 forms. The data output is performed through 32 reports, maps, and a dashboard. Data is stored and processed in a geodatabase using Microsoft SQL Server.

**Figure 6 ijerph-20-03282-f006:**
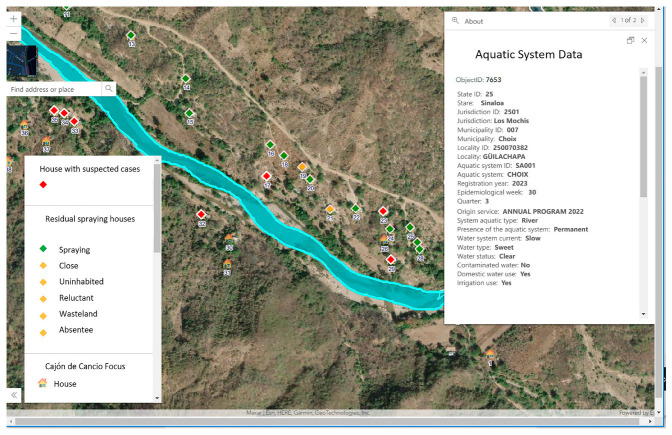
Geographic information system. Here is an example of a query, where the first layer contains epidemiological surveillance data, the second layer contains vector control data and the third layer shows the houses, these data layers are referred to the houses, that is why the unique identifier is shown in each of them. In addition, in the geographic viewer you can see the characteristics of the aquatic system.

**Figure 7 ijerph-20-03282-f007:**
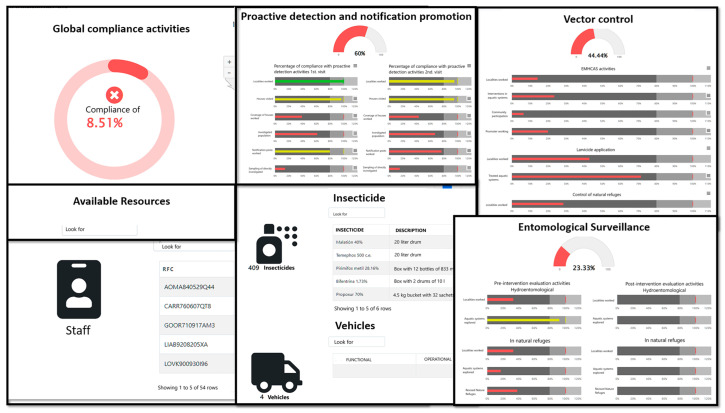
Dashboard. Indicators for monitoring the activity of notification, detection, promotion, vector control, and entomological surveillance. The dashboard indicators are calculated using data that the jurisdictional office routinely enters into the system. The categories of compliance of the activities are divided into Not carried out when it is 0%, Insufficient from 1 to 79.9% (red color), Acceptable from 80 to 99.9% (yellow color) and Optimum from 100% (green color).

**Figure 8 ijerph-20-03282-f008:**
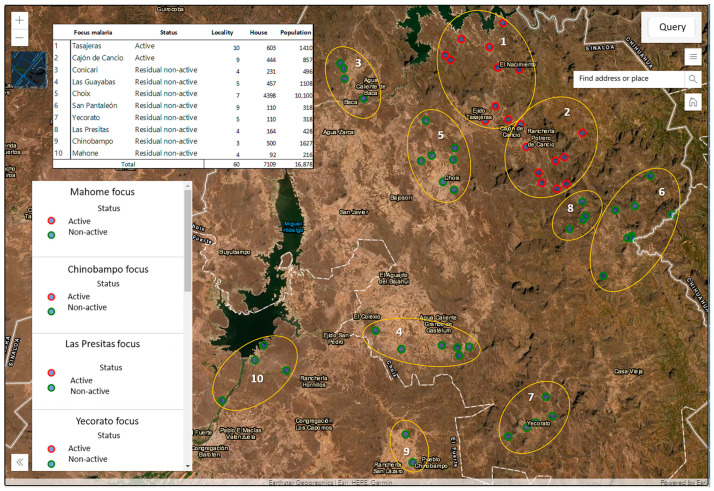
Geographic information system (spatial data viewer). Example of active and residual foci in the State of Sinaloa (yellow circles) identified with a consecutive number. You can see the localities where one or more autochthonous malaria cases were detected (within the last year) that were classified as active foci and the localities where malaria transmission was recently interrupted (between 1 to 3 years) that were classified as foci residuals.

## Data Availability

Not applicable.

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
