# Peer review of "Web Geographic Information System: A Support Tool for the Study, Evaluation, and Monitoring of Foci of Malaria Transmission in Mexico"

_ijerph, 2023, doi:10.3390/ijerph20043282_

Round 1
Reviewer 1 Report
ijerph-2075683-peer-review-comments-v1
Article
Web Geographic Information System for the Elimination of Malaria; Tools for the study, evaluation and monitoring of foci 3 of malaria transmission.
René Santos-Luna et.al.
Overall Recommendation :
Consider for publication after major revision.
Comments to the authors:
The development of a web-based GIS system for malaria elimination is highly desirable for any country to monitor the situation in real time for immediate evidence-based decisions and actions. Characterization, monitoring and investigation of foci is equally important in the elimination settings to clear the transmission foci. Prevention of re-establishment of transmission is expected to maintain the malaria free status after achieving malaria elimination. The work presented in this paper is therefore appropriate as well as timely considering the malaria elimination target of Mexico by 2025. The GIS system has been developed as per the resource availability in Mexico to overcome the inherent challenges that are local and focal to the country and areas where malaria is prevalent. This is a very good beginning and hopefully, the tool would be further refined and fine-tuned to make it more productive for the country.
As far as the presentation/writing part is concerned, the authors need to revise the entire paper right from the title to the conclusion due to the following reasons :
1. Language and grammar need extensive revisions for correctness as well as clarity.
2. The sentences are long with too many punctuation marks, this needs correction. All long sentences in the paper should be broken into smaller and simpler sentences.
3. Punctuation needs specific consideration right from the title to the conclusion. The excessive use of semi-colons and commas should be avoided since the text becomes incomprehensible as well as grammatically incorrect due to this issue.
Examples :
i) Title: The authors may consider changing the title to ‘Geographic Information System: An effective tool for the study, evaluation and monitoring of foci of malaria transmission.
ii) Line no. 11-14: The sentence ‘The country joined the WHO's E-25 initiative for the elimination of Plasmodium; vivax, to achieve elimination and certification within the established period, it was necessary to have information to assist in the detection, investigation, elimination of transmission in the Foci, as well as in the timely treatment thereof’ can be revised as ‘ The country joined the WHO's E-25 initiative for the elimination of Plasmodium vivax to achieve elimination and certification within the established period. The necessity to have a web-based information system was, therefore, felt to assist in the detection, investigation and elimination of transmission in the foci as well as for the timely treatment of malaria positive cases.
iii) Similarly, many technical terms have not been written appropriately and need revisions to ensure that the terminology recognized and accepted internationally is used for easy understanding by the readers.
iv) Since this work has already been accomplished, the purpose, results, etc. should be in the past tense.
v) Please check that the figures and tables are correctly referred in the text .
Examples of some wrong terminology, words and sentences are given below for the reference of the authors :
· Line 23: Integrated
· Line 38-39: In recent decades, in Mexico, have been implemented several strategies and each has had a significant impact on the reduction of malaria cases.The correct sentence should be – In recent decades several strategies have been implemented in Mexico and each has had a significant impact on the reduction of malaria cases.
· Line Nos. 99,104,108,116,120,123-125,138,193(PDR??)227(Rapid detection test),242,244,245,275(whom ?) 278,279(applied ?),281(most positive evidence with 12 ?), 285 (integrated with ?),292(of the called ?),293 (On the map you can see ?) ,295-298,300-301 (disposal environments ?), 306 (will need be), 310-312,310-321,353 (secret ?)
Discussion & Conclusion:
Desirably, all references to compare with similar studies should be given in the discussion part including those given in the conclusion. It would be desirable to include the limitations of the study should also give the purpose of the study and the results presented.
In conclusion, references are not required. Instead, the conclusion paragraph can focus on the short outcome of this paper and its future scope of work.
References should be quoted at the end of the sentence preferably.
Author Response
Dear Reviewer:
The authors really appreciate and value the comments and suggestions.
R1, R2, R3. The authors made an effort to review all sections of the manuscript to improve the writing and to attend to the suggestions of all the reviewers. Finally, the manuscript was submitted to the extensive review service by the English Edition Department (certificate is attached).
R4. We consider it pertinent to change the title of the manuscript, "Web Geographic Information System: A Support Tool for the study, evaluation and monitoring of foci of malaria transmission in Mexico"
R5. The technical terms were revised and replaced by internationally standard terms.
R6. The sequential order of the figures, numbering and references in the text was corrected
R7. A section "limitations of the study" was added, within the Discussion section
R8 The content of the conclusions was completely modified, a brief summary of the relevant points of the project was included, mainly the development and implementation of GISTEM.

Reviewer 2 Report
They need to improve on the flow of the paper. Most sentences are incomplete or just hanging. The majority of the abbreviations are different from what they are in full It requires extensive editing and revision. A very difficult paper to read.
Author Response
Dear Reviewer:
The authors appreciate and value the sincerity of the comments.
R1. The authors made an effort to review all sections of the manuscript to improve the writing and to attend to the suggestions of all the reviewers. Finally, the manuscript was subjected to extensive review by the English Editing Department. (certificate is attached).

Reviewer 3 Report
Thank you for giving me this opportunity to read the manuscript entitled "Web Geographic Information System for the Elimination of Malaria; Tools for the study, evaluation and monitoring of foci of malaria transmission". The topic of this manuscript is interesting and would be a good contribution to this field. I think it could be considered for publication in IJERPH once the following issues are addressed.
1. Please replace the keywords that already appear in the manuscript's title with close synonyms or other keywords, which will also facilitate your paper being searched by potential readers.
2. "Scale" and "Compass" should be added to the map in Figure 1.
3. There are quite some statements without references in the Introduction section, and the authors should add some references to support these statements.
4. Some of the texts in Figures, for example, those in Fig. 4, need to be bigger to be read clearly.
5. Limitation section should be added as a sub-section to the Discussion.
6. Some grammatical errors exist in the manuscript. Therefore, critically reviewing the manuscript's language will improve its readability.
Author Response
Dear reviewer:
The authors thank you for your comments and suggestions.
R1. The keywords that will be included are: Entomological Surveillance, Vector Control, Dashboard.
R2. Edited map to include scale and north symbol.
R3. In the introduction section, figures were updated and missing references were added, new references were also added due to suggestions from another reviewer.
R4. In Figure 2, the text size was increased, I must mention that said figure was changed to number 4. At the suggestion of another Reviewer.
R5. Added "Limitations" section
R6. The manuscript was submitted to a professional English editing review by the publisher's recommended office. (certificate is attached)

Reviewer 4 Report
The efforts by the authors to develop a web GIS as a support tool for malaria elimination is appreciable. However, there are a few queries/ suggestions for clarity.
Web GIS as such is not a tool which can eliminate malaria, rather it can serve as a support tool. Therefore, the title of the MS may be modified somewhat like “Web Geographic Information System as a support tool for the study, evaluation and monitoring of foci of malaria transmission in Mexico’
In abstract (last line), the authors have mentioned that the foci manager tool decreased the cost of spatial data collection. Kindly clarify the basis behind this conclusion.
Fig 2: What is ‘Desing’? Is it designing?
Fig 5: What is the role of mapping aquatic systems in transmission reduction or malaria elimination?
Fig 6: There is typographic error- Acuatic. Should be aquatic.
Fig 7: Kindly clarify the difference between active and residual outbreaks. The text pasted on the figure is hazy, may be replaced.
Fig8: What were the activities under entomological surveillance? And how the inputs were incorporated into the modules, kindly elaborate.
Line 287: Of overall 11225 RDTs, only 12 cases of malaria were detected indicating very low endemicity. Only focus, i.e. Campeche was positive.
Line 343-346: How the dereferencing of houses and aquatic systems helped in detection, relationship with malaria transmission and extermination of foci.
Line 413: ‘SIEM as a vector control tool’, kindly elaborate the same.
Page 12, item No 4.2: As such implementation of the SIEM has not been accomplished, therefore it may be omitted from here as well as from Fig.2. Further, the work of Ma et al 2016, in the present context, may be discussed in detail.
In conclusion the authors have emphasized to improve the system by integrating mobile tracker for fever cases (Pal et al. 2021). Mobile based app are quite different from web based GIS system as the latter is comprehensive in terms of georeferencing of houses, aquatic bodies, foci of transmission etc. Therefore, the authors should clarify whether they advocate mobile based app or comprehensive web GIS system.
Author Response
Dear reviewer: The authors thank you for your comments and suggestions.
R1. The title of the manuscript was modified, we found the title you propose to be very accurate.
R2. The summary sentence was modified, we clarified which tool contributes so that the georeferencing process reduces the cost.
R3. Changed the "Designing" text in Figure 2
R4. In the Aquatic Sites near the town with active transmission, the vector control office carries out entomological surveillance activities to apply mosquito control actions.
R5. Fixed the "Aquatic" text in the figure.
R6. Increased the text size of the figure. The difference between active and residual outbreak was clarified in the sentence of the text of figure 7
R7. The Control Panel figure was modified, one of the Emtomological Surveillance screens was added, a sentence was included to mention how the data is incorporated from its respective module.
R8. The paragraph was modified, the rapid tests have only been applied in four states, the percentage of positivity is very low.
R.9 The paragraph of that section was modified, georeferencing was a process to visualize the information in an integrated way, the information is what really helps the process of monitoring indicators of the strategy.
R10. Removed this paragraph, at the suggestion of another reviewer
R11. section was removed.R12. In the scope section, a paragraph was included to describe a possible interaction.
Reviewer 5 Report
Congratulations to the authors. Although the creation of a viewer is not something new or innovative. The job is well done. But I think that the authors should improve the references with antecedents of international impact and journals such as Geo-information MDPI. As well as proposing a plan of future actions that will be carried out with this viewer. Well, the important thing would be to combine the information with social, economic, health indicators, etc. So that the viewer is an important tool.Author Response
Dear reviewer:
The authors thank you for your comments and suggestions.
In the introduction section, references were added, including one from the Geo-Information journal, where they also applied geospatial technology to develop a support tool in the Malaria Elimination process.
A possible action plan was also added in a new "Scope" section, in which the following steps are mentioned.
Round 2
Reviewer 1 Report
The authors have tried revising the paper and succeeded to some extent. However, major concerns remain as the paper has still not been appropriately revised. Several issues continue to exist and to resolve those, following suggestions might be useful for further improvisation :
1. Reorganize the contents of the paper under the headings to ensure that i) Introduction contains information regarding the study and what is already known/published on the topic so far ii) Material & methods contains a description of this particular study only and all relevant figures and graphs explaining the methodology are included under this heading iii) Results cover the appropriate details i.e. outcome of this study along with the relevant figures and tables etc. iv) Discussion should align with the details given under the introduction and results. Discussion should also cover the results included in the paper in comparison to what has already been published and try to bring out similarities, and dissimilarities as well as what new has been presented in this paper. limitations of the study, if any, should be a part of the discussion. The scope should deal with the future of the study in light of the objectives and conclusions drawn in this paper. The conclusion should contain the outcome of this study only.
2. It is desirable to write short and clear sentences for clarity and correctness.
3. Nomenclature and abbreviations accepted and used internationally should not be changed e.g., DDT, Rapid Diagnostic Test etc.
4. Please ensure that the references are quoted appropriately and in order as per the revised paper.
It is beyond the purview and scope of a reviewer to correct and suggest changes in every sentence. For better comprehension and to facilitate further revision of this paper some suggestions have been made on the revised version of the manuscript itself and uploaded herewith. It is expected that taking a cue from these suggestions, the authors will be able to revise the entire paper accordingly to make it fit for publication.

Author Response
Dear Revisor,
Thanks for your comments.
R1. All sections were reorganized.
i) The introduction was reviewed and modified.
ii) Material and methods were rewritten according with suggestions.
iii) Results were rewritten according with suggestions.
iv) Discussion was rewritten according with suggestions, and we included compare with another systems.
R2. The entire document was revised and rewritten to avoid long sentences.
R3. The nomenclature was revised so that it was correctly.
R4. The references were revised so that it was correctly.
We reviewed your suggestions and corrected everything that was wrong.
Reviewer 2 Report
A much improved and clearer presentation. Can easily be followed and is worth publishing
Author Response
Dear Revisor,
Thanks...
Reviewer 3 Report
Thank you for allowing me to read the revised version of the manuscript titled "Web Geographic Information System: A Support Tool for the study, evaluation and monitoring of foci of malaria transmission in Mexico", and for the detailed responses to my earlier comments. I am satisfied with this revised version, and I think it is acceptable now.
Author Response
Dear Revisor,
Thanks...
Reviewer 4 Report
The authors have addressed almost all the comments satisfactorily.
The text in the Fig 1 to 8 is not legible, therefore all the figures need replacement with clearly visible text.
Under keyword, 'malaria' and 'GIS' may also be added.
Author Response
Dear Revisor:
Thanks for your comments.
R1. Resolution increased to 600 DPI for all figures.
Part of the title was modified in all the figures, from figure 3 the format and explanatory text were modified, now the title can be distinguished from the text of the additional explanation.
R2. The keywords suggested were added.